# An Uplink Channel Estimator Using a Dedicated Instruction Set for 5G Small Cells

**DOI:** 10.3390/s22030753

**Published:** 2022-01-19

**Authors:** Biao Long, Dake Liu, Yipeng Sun

**Affiliations:** State Key Laboratory of Marine Resource Utilization in South China Sea, School of Information and Communication Engineering, Hainan University, Haikou 570228, China; longbiao@hainanu.edu.cn (B.L.); 19085208210029@hainanu.edu.cn (Y.S.)

**Keywords:** 5G small cell, channel estimator, Wiener filter interpolation, dedicated instruction set acceleration

## Abstract

As 5G small cells gradually become the main force of 5G indoor deployment, it is necessary to study channel estimators for 5G small base stations, but there has been limited research on high-performance channel estimators in recent years. This study implemented a low-delay, low-overhead, relatively universal channel estimation module by dedicated instruction set acceleration including reference signal estimation, Wiener, 1st order, and 2nd order interpolations in frequency and time domains. The instruction level acceleration is on our vector processor, yet is suitable for other commercial and academic vector processors. Through instruction acceleration, compared with the existing general vector processing instruction sets, the processor performance of the LS estimation module and Wiener filter interpolation in the frequency domain is improved by 50% and 37.5%, respectively. The BER VS SNR measure of time–frequency Wiener filter interpolation achieves 4 db compared with linear interpolation, meaning our instruction level acceleration can be an optimum solution.

## 1. Introduction

The fifth generation mobile communication technology, 5G, has higher speed, lower delay and denser connections than 4G. With the gradual development of commercial deployment of 5G networks, 5G small cells built to supplement the radiation blind area of the macrocell [1,2,3] are becoming more popular. In the macrostation, it is a common practice to use an application-specific integrated circuit (ASIC) as the solution for the base station hardware module. Different from the 5G massive MIMO supported by outdoor macrocells, small indoor cells generally have a small number of antennas in order to meet low power consumption and low radiation requirements. Therefore, small cells are still suitable for small-dimensional MIMO. In recent years, there have been some studies on channel estimation of 5G small-dimensional MIMO using FPGA [4]. Because of the high power consumption brought by the FPGA solution, the scheme of using ASICs in small cells is subsequently proposed. In the new 5G technology environment, receivers need to cope with changing channel configurations and functional requirements. Therefore, relatively flexible channel estimators and equalizers designed based on application-specific instruction set processors (ASIPs) are replacing traditional ASICs. In the 4G era, there were studies on channel estimation using ASIPs [5]. ASIPs implement domain-specific work by setting a dedicated instruction set. Compared with FPGA and general-purpose processors, ASIPs further reduce the construction cost and power consumption of small cells, and also adapt to the iteration of 5G versions in software maintenance. This is due to the flexibility of ASIPs in specific areas and the silicon efficiency brought about by the reuse of ASIP hardware [6].

In recent years, there have been many studies on channel estimation. For example, reference [7] implements a multimode decision-oriented channel estimation ASIC for 2 × 2 MIMO-OFDM, which completes the work of using ASICs to realize small-dimensional MIMO in the early years. A software transceiver for a 5G new radio physical uplink shared channel (PUSCH) is proposed in reference [8]. It completes the software implementation of 5G PUSCH of 2 × 2 MIMO, but lacks the discussion of hardware implementation. A 5G/6G channel estimation vector digital signal processor based on software configuration for dynamic switching between delay and throughput efficient algorithms is proposed in reference [9]. In order to solve the problem that it is inconvenient to change the hardware algorithm and configuration on the vector processor, reference [9] chooses the software implementation to realize the dynamic switch between the algorithm and configuration. In recent years, there have been many studies on 5G MIMO based on machine learning. For example, references [10,11,12] use the theory of machine learning to study channel estimation in MIMO systems. The traditional algorithm can be optimized by machine learning, and the result is often better than the original algorithm. However, the convergence time of machine learning is long, which means that it is difficult to meet the requirements of real time and low power consumption of 5G small cells.

To improve small cell MIMO capability, this study used the method based on ASIPs to implement the channel estimation module of 4 × 4 MIMO by developing a dedicated instruction set on the vector processor (VP). This study designed a set of dedicated processor instructions, which could not only run the channel estimation function on the VP, but also use some of the instructions for channel equalization, signal detection and other necessary functional modules in 5G uplink. Using different instruction combinations supported by our dedicated instruction set, different algorithms and functions can be realized with a relatively small number of hardware units. Using the design method of ASIPs, we can obtain higher flexibility than ASICs [13], leaving a design margin for the development of subsequent base station algorithms and extending the life cycle of the product. At the same time, compared with the CPU based on the general instruction set, the ASIP has lower power consumption and a smaller hardware area [14], which can reduce the construction cost of small cells.

In fact, the utilization of assembly instructions also follows the Pareto rule. Twenty precent of commonly used instructions account for about 80% of the running time of small cell processors, so 20% of instructions must be accelerated [15]. Therefore, the main innovation of this paper is to accelerate the instruction set used by ASIPs. Compared with the general vector processor instruction set, it can complete the channel estimation with fewer instructions. The main tasks involved in this study are: at the transmitting end, the demodulation reference signal (DMRS) is mapped to the OFDM resource block according to the 3GPP standard; at the receiving end, the channel estimation algorithm of the pilot point is completed by designing a simple LS estimation algorithm; a dedicated instruction set is designed to use the VP for channel estimation of pilot points and interpolation estimation of non-pilot points; the instructions in the subject loop of the channel estimation module are merged and accelerated, which speeds up the calculation of the processor; finally, the instruction overhead estimates of linear, second-order and Wiener filter interpolation in the time–frequency domain are summarized.

The rest of the paper is organized as follows. Section 2 introduces the basic model of 5G small cells, including the process of signal sending and receiving and the standard of pilot design. Section 3 introduces the main algorithms used and the methodology of designing a dedicated instruction set and acceleration for the algorithm. Section 4 shows the pseudo code of the algorithm, the assembly code of the instruction and the optimization effect after acceleration. Section 5 contains the conclusion and the significance of the research.

## 2. Basic Model

### 2.1. 5G Uplink Channel Transceiver Process

After LDPC coding and rate matching, the processing of the baseband transmitter chain at the physical layer level includes scrambling code, modulation symbol mapping, DMRS generation, precoding, etc. These processes are described in detail in the Section 6.3.1 of the 3GPP TS 38.211 [16] standard document on the 5G uplink physical layer. The subcarrier mapping modulation symbol data are OFDMA modulated by an IFFT operation. The signal modulated by OFDMA is transmitted by the transmitter and received after transmission in the wireless channel. The receiver redivides the OFDMA data into data of multiple subcarriers through FFT operation [17], and estimates the channel according to the known data (pilot) inserted at the transmitting end. Finally, all the transmitted data are inversely derived according to the results of channel estimation. Combined with the process of channel estimation at the receiver, the basic flow of 5G uplink can be summarized as shown in Figure 1 [16]. The main work of this paper is in the channel estimation module. A detailed flowchart can be found in Section 3.

### 2.2. Pilot Resource Mapping Model

The main functions of uplink physical channel include correlation demodulation and channel measurement. In PUSCH, DMRS is used for channel estimation. In 5G uplink, the mapping mode of PUSCH is star pilot, which inserts pilots in frequency domain subcarriers and time domain symbols at standard intervals. The work of this section is part of the resource mapping module in Figure 1. After resource demapping, the channel can be estimated according to the distortion of the DMRS between the transmitter and the receiver.

For the frequency domain DMRS mapping method, 3GPP TS 38.211 indicates the mapping position of the DMRS in the PUSCH. The research of this paper is based on the channel estimation of 4 × 4 MIMO 5G small cells, so the data are divided into four layers for transmission. In addition, small cells serve pedestrians or devices in the Internet of Things, which have low mobility, so the single-symbol DMRS meets the requirements. Combined with the standard 38211, the DMRS frequency domain mapping method used in this paper is shown in Figure 2. The structure of the 12 subcarriers * 14 symbols is called a resource block (RB), which is a basic unit of resource scheduling. A time–frequency resource unit consisting of an OFDM symbol and a subcarrier is called a resource element (RE). The DMRS is mapped on the RE agreed by both the sender and the receiver in the way shown in Figure 2.

## 3. Algorithm and Implementation of Channel Estimation

### 3.1. The Methodology of Designing a Dedicated Instruction Set

After selecting the use of a dedicated instruction set as the research method of channel estimation in this study, it follows a flow chart [18] which is suitable for the design of the dedicated instruction set, as shown in Figure 3.

The first step is demand analysis. The performance of the wireless communication system is largely affected by the wireless channel, which makes the propagation path between the transmitter and the receiver very complex. The wireless channel is very random, so the receiver needs to constantly evaluate and predict the channel environment. For example, the receiver estimates the channel response of the entire subspectrum using known signals (pilots) distributed on different subspectra. The basic process of pilot-based channel estimation is to estimate the channel at the pilot first, then obtain the channel estimation between pilots by interpolation or filtering. Finally, the channel estimation of the whole channel is used for channel equalization to eliminate the distortion of the received data.

After understanding the requirements and basic methods of channel estimation, we can choose the algorithms with low cost and good enough performance as the basis of the follow-up work. The selected algorithms are shown in Section 3.2 and Section 3.3. After selecting the algorithms, we need to test the function and performance of the algorithms. The functional verification and performance of the algorithms are shown in Section 3.4. After the test is passed, we need to analyze the code in order to have a comprehensive understanding of the instruction set that needs to be designed and analyze the possibility of code acceleration. The work of code analysis is shown in Section 4.2 and Section 4.3. The next step is to design a dedicated instruction set that contains accelerated instructions, as well as the arrangement and verification of assembly instructions, which can be seen in Section 4.3, Section 4.4 and Section 4.5. After designing the instruction set and assembly code and verifying them, the final work is to carry out the microarchitecture design of the hardware module and gate-level circuit design to realize the hardware function according to the previous research. The research on hardware design is not in the scope of this paper, but it relates to the goal of the follow-up research of this paper.

### 3.2. Channel Estimation at Pilot

In 5G communication, both the sender and the receiver agree on the known demodulation reference signal (DMRS) for both sides to measure the distortion and interference of the OFDM subspectrum to obtain the channel response values at the reference point. On this basis, the channel response values between pilots are obtained by interpolating values between two or three reference signals. Interpolated values will be used as the channel estimation for received signals. Therefore, this study used the pilot-based OFDM channel estimation method. In the channel estimation at the DMRS, the LS [19] + Wiener interpolation [20] method is suitable for small cells due to its complexity and performance trade-off, and it is widely used in OFDM channel estimation.

Consider four users. For a certain subcarrier, let *si* be the data sent by the *i*-th user, *hij* is the channel from the *i*-th transmitting antenna to the *j*-th receiving antenna and *yj* is the channel of the *j*-th receiving antenna. Then, the received value obtained on each antenna is shown in Equation (Equation 1) [21]. Each of the *si***hij* represents the reception component of the *i*-th transmitting antenna on the *j*-th receiving antenna after passing through its wireless channel. Each receiving antenna has four components from four different transmitting antennas. The transmission model corresponding to Equation (Equation 1) is shown in Figure 4.
(1)$$\left\{\begin{array}{c} s_{1}*h_{11}+s_{2}*h_{21}+s_{3}*h_{31}+s_{4}*h_{41}+n_{0}=y_{1}\\ s_{1}*h_{12}+s_{2}*h_{22}+s_{3}*h_{32}+s_{4}*h_{42}+n_{0}=y_{2}\\ s_{1}*h_{13}+s_{2}*h_{23}+s_{3}*h_{33}+s_{4}*h_{43}+n_{0}=y_{3}\\ s_{1}*h_{14}+s_{2}*h_{24}+s_{3}*h_{34}+s_{4}*h_{44}+n_{0}=y_{4} \end{array}\right.$$

It can be seen from the DMRS structure of Figure 2 that for each subcarrier, only two users send the DMRS, and the other two users do not send data. Taking DMRS ports 0 and 1 as an example, user 3 and user 4 do not send data that the corresponding *s*_3_ and *s*_4_ are both 0, so Equation (Equation 1) can be simplified to Equation (Equation 2).
(2)$$\left\{\begin{array}{c} s_{1}*h_{11}+s_{2}*h_{21}+n_{0}=y_{1}\\ s_{1}*h_{12}+s_{2}*h_{22}+n_{0}=y_{2}\\ s_{1}*h_{13}+s_{2}*h_{23}+n_{0}=y_{3}\\ s_{1}*h_{14}+s_{2}*h_{24}+n_{0}=y_{4} \end{array}\right.$$

As shown in Equation (Equation 2), the LS channel estimation at the pilot cannot be in the form of a simple Y/S. In order to calculate Equation (Equation 2), we can choose to divide two adjacent DMRSs on the same port into a CDM group in the time domain or frequency domain, and assume that the H value of the same CDM group is the same. Then, the two pilot symbols of the same CDM group can be combined with equations to obtain the estimation result of LS estimation with the influence of noise.

The work of filtering the noise can be placed in the Wiener filter interpolation module between the DMRS and the MMSE channel equalization module. The method adopted in this study is to divide two adjacent DMRSs on the same port into a CDM group in the time domain. In this way, an accurate solution can be obtained in the frequency domain, and an approximate solution can be obtained in the time domain. The loss of accuracy in the time domain is negligible in low-mobility small cell scenarios. If the mobility increases, more DMRSs can be inserted in the time domain to solve the problem of decreased estimation performance due to increased mobility.

### 3.3. Interpolation Estimation between Pilots

From Figure 2, after obtaining the channel response value at the DMRS, for each port, there are still many channel response values of the RE in time domain direction and frequency domain direction that are not known. Therefore, we need to fill the channel response value of the whole RB by interpolation. Interpolation between subcarriers is called frequency domain interpolation, and interpolation between symbols is called time domain interpolation. The common channel estimation interpolation algorithms between pilots are linear interpolation, second-order interpolation, Wiener filter interpolation, etc. The 1st-order interpolation (linear interpolation) algorithm is a relatively simple interpolation algorithm, which uses the channel estimation values of the two adjacent pilot points to interpolate the channel response values of the data between the two pilot points. The 2nd-order interpolation algorithm uses the channel response values of the three adjacent pilot points for interpolation. Wiener interpolation is a kind of interpolation algorithm based on the MMSE criterion, and its specific method is described in the next part of this paper. Among them, the linear interpolation algorithm is the simplest, and the performance of second-order interpolation may be better than that of linear interpolation when the channel environment changes sharply. As the noise cannot be eliminated by using LS estimation for pilots, Wiener filter interpolation is used in the channel estimation between pilots to reduce the impact of noise on the overall channel estimation performance. The Wiener filter interpolation algorithm needs to obtain prior information of the channel in advance, such as the channel delay estimation parameters, to calculate the correlation matrix. However, in actual situations, it is difficult for PUSCH to obtain this channel prior information before channel estimation. The usual approach is to pre-store the autocorrelation and cross-correlation matrix items of some commonly used statistical channel model scenarios, or even directly store some typical estimator coefficients. The device detects the actual channel scene and matches the closest LMMSE estimator coefficient for use. Taking into account the calculation efficiency and performance, linear interpolation between DMRSs is also used. One of the innovations of this paper is that linear interpolation can be made into the form of taps, and linear interpolation and Wiener filter interpolation can be integrated into one algorithm and architecture on hardware instructions.

The formulas of frequency domain and time domain Wiener filtering are expressed by Equation (Equation 3) [22], where *R_HP_* is the cross-correlation matrix, *R_PP_* is the autocorrelation matrix, β is the modulation coefficient and *I_N_* means identity matrix. This means that in the process of calculating the filtering coefficient *W*, it is necessary to invert the autocorrelation matrix *R_PP_* after adding noise.
(3)$$W=R_{HP}{\left(R_{PP}+I_{N}*\frac{\beta }{SNR}\right)}^{-1}$$

In reference [23], a method for calculating the correlation coefficient of the frequency domain matrix using the prior information of the channel rather than the specific values of the transmitting and receiving antennas on each subcarrier is given, as shown in Equation (Equation 4). The complexity of this algorithm is low, and the prior information of channels in different scenarios has been measured and disclosed by various standard organizations. For example, in Section 3.4, the multipath model provided by the 3GPP standards organization for pedestrians is shown. The channel parameters can be stored in the hardware in advance by the look-up table.
(4)$$r_{corf}\left(\Delta k\right)=\sin c\left(\pi \tau_{c\max}\Delta kf_{sub}\right)$$

From the above equation, we need to know the prior information of the channel as follows: τ_*c*max_ is the maximum delay spread of the multipath channel; fsub is subcarrier frequency; Δ*k* is the frequency domain subcarrier spacing. We worked with a company to determine the parameters of Wiener filter interpolation.

### 3.4. Algorithm Simulation Performance

According to the pilot structure recommended by 3GPP-38211 described in Section 2, this paper proposes channel estimation algorithms in Section 3. Using the multipath Extended Pedestrian A (EPA) model proposed by 3GPP-36101 [24], the LS estimation + Wiener filter interpolation algorithm was simulated. The number of OFDM subcarriers set in the simulation in this paper is 1024 (assuming it is a user’s PUSCH channel). EPA is an extended pedestrian channel proposed by 3GPP. Under the EPA channel model, the UE moves slowly, so the Doppler spread is small. It is a simulation environment model that conforms to the small cell scenario. According to the 3GPP protocol, the Doppler of EPA is extended to 5 Hz, and the power delay data of its seven multipaths are shown in Table 1.

For the parameters of Equation (Equation 4), according to the EPA channel model, the maximum delay spread τ_*cmax*_ of the multipath channel is set to 410 ns and its reciprocal corresponds to the coherent bandwidth, which is an important parameter to characterize the constant gain and linear phase of the multipath channel. The subcarrier spacing *fsub* is 15 kHZ; according to Figure 2, in the autocorrelation matrix, Δ*k* = 2; in the cross-correlation matrix, Δ*k* = 1. In order to simulate the channel environment of small cells, we set the antenna correlation to low correlation. The BER curves of time–frequency Wiener filter interpolation and time–frequency linear interpolation under different modulation modes are shown in Figure 5a. The BER curve obtained by the combination of different interpolation methods under 64 QAM is shown in Figure 5b.

In Figure 5a,b, “F -” indicates the interpolation method used in the frequency domain; “T -” represents the interpolation method used in the time domain; “one” refers to linear interpolation; “second” refers to second-order interpolation; and “wiener” refers to Wiener filter interpolation. From Figure 5a, when the SNR increases gradually, the effect of Wiener filtering in the time–frequency domain is obviously better than that of linear interpolation, and the overall improvement effect is between 0 and 4 db. From Figure 5b, the use of frequency domain Wiener and time domain Wiener can each bring about a 2 db gain compared with linear interpolation. However, when the SNR is high, the BNR of second-order interpolation is worse than that of linear interpolation. This is because there is only one subcarrier between the two pilots in the frequency domain, so second-order interpolation does not have an advantage over linear interpolation; in the time domain, the mobility of the EPA model is low, the channel changes slowly in the time domain and linear interpolation is more suitable than second-order interpolation.

## 4. Design and Acceleration of Instruction Set

### 4.1. Vector Processor Instruction Set Architecture

According to 5G micro-base stations, the typical feature of the baseband algorithm is its predictable small code and data volume. The team in our laboratory has developed a set of very long instruction word (VLIW) vector processing for baseband processing MCU + load/store + single instruction multiple data (SIMD). The specific architecture is described in [25]. The processor platform is based on an SIMD kernel design with variable data-level parallelism. For the convenience of scientific research, the native data width of the SIMD core is a 16b + 16b complex variable. Its framework includes basic time domain and frequency domain vector, matrix, transform, filtering and power and noise estimation instructions for baseband symbol processing. However, this vector processor still lacks commonly used acceleration instructions for specific operations, such as Weiner filtering, linear interpolation and matrix inversion. The work of this paper is based on the Wiener interpolation-oriented instruction set acceleration of the VP.

### 4.2. Pseudo Code Analysis of Algorithm

From the discussion in Section 3, we know that for a certain pilot subcarrier, the pilot value of the transmitting antenna i1 is *R*1, and the pilot value of the transmitting antenna i2 is *R*2; the receiving value of the fourth symbol on the receiving antenna j1 is *Z*1, and the receiving value of the eleventh symbol is *Z*2; the receiving value of the fourth symbol on the receiving antenna j2 is *Z*3, and the receiving value of the eleventh symbol is *Z*4. The corresponding estimated value of the channel *H_i_*_1_*_j_*, *H_i_*_2_*_j_* can be obtained, and the calculation formula can be found in Equations (5) and (6), which are rewritten according to Equation (Equation 2).
(5)$$H_{i1j}=\left(Z1^{*}R2+Z2^{*}R1\right)/\left(2^{*}R1^{*}R2\right);$$
(6)$$H_{i2j}=\left(Z1^{*}R2-Z2^{*}R1\right)/\left(2^{*}R1^{*}R2\right);$$

It is easy to find that the LS estimation of multiple subcarriers inevitably requires multiple cyclic Equations (5) and (6). Therefore, we need to use “For” to carry out multiple loops in the pseudo code in Algorithm 1. In order to reduce the clock cycle of LS operation and reduce the overhead of pipeline emptying and memory access due to multiple cycles, we need to carry out loop unroll. Pipeline technology can be used as efficiently as possible. According to the capability of the existing SIMD architecture, and in order to eliminate the NOP inserted due to data dependency as much as possible, we need multipoint parallel loop unroll for multiple subcarriers. The pseudo code implementation is shown in Algorithm 1.
**Algorithm 1** LS estimation algorithm.**Input:** 
Z1[4 * N], Z2[4 * N], R1[4 * N], R2[4 * N] //4 * N pilot subcarriers;**Output:** 
Hi1j[4 * N], Hi2j[4 * N]1:**for**i=0→N**do**2:    **for** j=4→N **do**3:        //Calculate 4 subcarriers in one cycle4:        a[4 * i + j] = Z1[4 * i + j] * R2[4 * i + j]+Z2[4 * i + j] * R1[4 * i + j]; //Subcarrier 4i + j5:        b[4 * i + j] = 2 * R1[4 * i + j] * R2[4 * i + j];6:        c[4 * i + j] = Z1[4 * i + j] * R2[4 * i + j]−Z2[4 * i + j] * R1[4 * i + j];7:        Hi1j[4 * i + j] = a[4 * i + j] / b[4 * i + j];8:        Hi2j[4 * i + j] = c[4 * i + j] / b[4 * i + j];9:    **end for**10:**end for**

The calculation method of the Wiener filter interpolation algorithm is shown in Equation (Equation 3). Taking the pilot configuration of 3GPP TS 38.211 introduced in Section 2 as an example, the interpolation formula of the data subcarrier to be interpolated between the two pilots is shown in Equation (Equation 7), where *H_ls_*_1_, *H_ls_*_2_ is the H matrix of the first pilot subcarrier and the second pilot subcarrier, respectively; *W*1, *W*2 is the element of the corresponding tap matrix; *H*0 is the H matrix of the subcarrier to be interpolated. It is easy to find that the interpolation formula of linear interpolation can also be transformed into a form similar to Equation (Equation 7).
(7)$$H0=W1^{*}H_{ls1}+W2*H_{ls2};$$

For linear interpolation, the tap value only needs to be made into a look-up table, and different tap values can be selected according to the pilot interval. For Wiener interpolation, the correlation matrix is generated by the sinc function, and the specific element values can be calculated in advance and put into the look-up table. Before the interpolation module starts to work, the value of the Wiener filter correlation matrix and the SNR in the configuration register are read into the register file, and then called by the operation module.

The pseudo code of the interpolation module is shown in Algorithm 2.
**Algorithm 2** Wiener interpolation module in frequency domain.**Input:** 
RPP[2 , 2], RHP[2 , N], n * Hlsk[2 , 4] //N is the pilot interval, and the number of pilot subcarriers is 2N, and k is the sequence number of pilot subcarriers (k = 1,2,3…,2 * n means interpolation, k = 0,2 * n + 1 means extrapolation).**Output:** 
n * (N + 1) * H[4 * 4]1:Hls0[2 , 4] = 2 * Hls1[2 , 4]−Hls2[2 , 4]; //The first and last virtual pilot subcarriers are obtained by linear extrapolation.2:Hlsn+1[2 , 4] = 2*Hlsn[2 , 4]−Hlsn-1[2 , 4];3:**for** 
i=0→n 
**do**4:    // i is addressing H at the pilot5:    **if** way==0 **then**6:        LUT RPP,RHP,β/SNR;7:        RPP’ = RPP + β/SNR*IN;8:        iRPP’ = Inverse (RPP’);9:        W = RHP * RPP’;10:    **else**11:        LUT W;12:    **end if**13:    **for** i=0→n **do**14:        //j is to address the H between pilots15:        Hj,Uhalf[2 , 4] = W1n * Hls2(i-1)[2 , 4] + W2n * Hls2i[2 , 4]; //the interpolation of the upper half H matrix16:        Hj,Lhalf[2 , 4] = W1n * Hls2i-1[2 , 4] + W2n * Hls2i+1[2 , 4];//the interpolation of the lower half H matrix17:    **end for**18:**end for**

### 4.3. Instruction Set Design and Acceleration Opportunities

#### 4.3.1. Micromanipulation Decomposition and SIMD Data Parallel Instructions

For the LS module, first of all, we can make full use of the feature that an SIMD instruction in VLIW architecture can take multiple data for instruction optimization. In order to facilitate the discussion, we first discuss the micromanipulation decomposition corresponding to pseudo code with a small degree of parallelism. Setting the parameters as described in Section 4.2, we obtain Equations (8)–(11).
(8)$$H_{i1j1}=\left(Z1^{*}R2+Z2^{*}R1\right)/\left(2^{*}R1^{*}R2\right)$$
(9)$$H_{i2j1}=\left(Z1^{*}R2-Z2^{*}R1\right)/\left(2^{*}R1^{*}R2\right)$$
(10)$$H_{i1j2}=\left(Z3^{*}R2+Z4^{*}R1\right)/\left(2^{*}R1^{*}R2\right)$$
(11)$$H_{i2j2}=\left(Z3^{*}R2-Z4^{*}R1\right)/\left(2^{*}R1^{*}R2\right)$$

For Equations (8)–(11), the microoperations we need to carry out are as follows:①calculate B = 2**R*1**R*2;//2 clock acceleration:clock1 times clock2 shift; including operand forwarding;②calculate C1 = *Z*1**R*2 + *Z*2**R*1, C2 = *Z*1**R*2 − *Z*2**R*1;③calculate OOX = 1/B;//numerator and denominator of real number in clock1, reciprocal by LUT in clock2, reciprocal in clock3 using multiplication; including operand forwarding;④calculate C3 = *Z*3**R*2 + *Z*4**R*1, C4 = *Z*3**R*2 − *Z*4**R*1;//including operand forwarding;⑤NOP //C3 & C4 need two execution cycles, inserted NOP are for result waiting;⑥calculate *H_i_*_1_*_j_*_1_ = C1*OOX, *H_i_*_2_*_j_*_1_ = C2*OOX, *H_i_*_1_*_j_*_2_ = C3*OOX, *H_i_*_2_*_j_*_2_ = C4*OOX.

In normal DSP architecture, MAC-like instructions need two execution cycles, and results are not available right after the first execution cycle. Results from other instructions are available at the end of the single execution cycle thanks to the register forwarding technique.

In order to eliminate the NOP inserted in step 5, we need to parallel calculate the unrelated subcarrier data in the instruction pipeline. After a plurality of subcarriers are calculated respectively in step 4, the next step in the pipeline can be calculated without inserting the NOP.

Taking the Wiener filter interpolation module in the frequency domain as an example, in order to implement Equation (Equation 3) and (4), the microoperations that need to be carried out in the loop are as follows:①calculate RPP’ = RPP++β/SNR*IN;//autocorrelation matrix RPP add SNR diagonal matrix;②calculate the reciprocal ad-bc of the determinant of RPP’: [a b; c d] //including operand forwarding;③calculate the cofactor of RPP’:[d −b;−c a];④calculate OOY = 1/(ad−bc)//reciprocal of real numbers; including operand forwarding;⑤calculate the inverse matrix iRPP’ = OOY*[d −b;−c a];//including operand forwarding;⑥calculate W = RHP*iRPP’;//including operand forwarding;⑦calculate Hj,Uhalf[2, 4] = W1n*Hls2(i−1)[2, 4] + W2n*Hls2i[2, 4];//8 H values of non-pilot points are obtained by interpolation;⑧calculate Hj,Lhalf[2, 4] = W1n*Hls2i-1[2, 4] + W2n*Hls2i + 1[2, 4];//The other 8 H values of non-pilot points are obtained by interpolation.

The above work is to decompose the algorithm module into microoperations according to pseudo code. After obtaining the micromanipulations, we need to carry out instruction subcarrier parallelism and data parallelism according to the SIMD capability of the processor. In this paper, subcarrier parallelism is to execute multisubcarrier data in parallel at the same step by an SIMD instruction. Data parallelism refers to the simultaneous processing of multidata or multisteps of data independence computation in a subcarrier by an SIMD instruction. Traditional general vector processors can only perform simple subcarrier parallel computing. The VP used in this paper can support data parallel SIMD instructions because of the use of special data channels. Since the design of this study is based on a VP with 512-bit SIMD, there are a maximum of eight complex multipliers (32 real integer multipliers), and the relatively general SIMD parallel data instructions are shown in Table 2.

In Table 2, instruction 1, 10 operates only on different data of the same subcarrier, so it is a data parallel instruction. Instructions 2–9 can carry out either data parallelism or subcarrier parallelism. In Section 4.3.2, the instruction 1, 2, 10 is used for data parallelism, while 3–9 are used for subcarrier parallelism.

#### 4.3.2. Instruction Fusion Acceleration Opportunity

The instruction fusion acceleration proposed in this paper refers to the packaging of micromanipulation sets that need to be accelerated without adding hardware. Accelerating a certain part of the channel estimation module means that the work that originally needs several instructions to be completed can be completed with only one instruction after acceleration. In this case, the overall hardware operation unit does not increase, but the number of times that data need to be read and written from the register file become less, which improves the operation speed of the module as a whole.

For example, the 2*R1*R2 in Equations (8)–(11) can be made into an instruction to multiply and shift two complex numbers. In this way, compared with a two-complex multiplication and a shift instruction, the instruction efficiency and performance are improved, and the number of multipliers is not increased. This instruction can also be used in other programs.

The complex division instruction of 1 complex number divided by 1 complex number is also an opportunity to accelerate instruction fusion. For the formula of complex division, see Equation (Equation 12). It can be seen that complex division is also composed of 2 complex vector multiplications and adding 2 complex vectors, 2 complex vector multiplications and subtracting 2 complex vectors and real number division. All these microoperations (instructions) can be merged into three instructions: the denominator is obtained from the plural table; the reciprocal of complex multiplication; the result of complex multiplication. The implementation of complex division before instruction fusion requires five instructions.
(12)$$\frac{a+bi}{c+di}=\frac{ac+bd}{c^{2}+d^{2}}+\frac{bc-ad}{c^{2}+d^{2}}i$$

In the interpolation module of calculating the tap, the H value of the subcarrier to be interpolated can be obtained by multiplying the H value of the tap and the pilot point by using the basic instruction of 2MM1. However, for each interpolation, the first pilot subcarrier currently used in interpolation is the last pilot subcarrier used in the last interpolation, and the general vector instruction set does not support such a data arrangement. Therefore, this study designs an interpolation instruction to reduce the interpolation work that originally needs two instructions to one, that is, to complete the acceleration of the interpolation module. The instruction fusion acceleration instructions designed in this study are shown in Table 3.

The LS estimation module can be completed by using instructions 1–2 and 13–14, and the Wiener filtering module in the time–frequency domain can be completed by using instructions 3–12. After calculating the corresponding coefficients in advance, instruction 8CM1C and instruction 8CM8C can be used for first-order and second-order interpolation in the time–frequency domain using accumulators. These 14 instructions are designed to realize the function of channel estimation on the basis of the original VP. Instructions 2–9 can find similar instructions in the existing vector general instruction set such as the Intel AVX-512 and Arm Neon instruction set. The instruction 1, 10–14 is an accelerated instruction specially put forward in this paper to improve the efficiency.

### 4.4. Assembly Instruction Arrangement

In order to improve the pipeline efficiency of the loop, we need to carry out loop unroll. By using the multisubcarrier operation, the NOP instructions inserted because of the data dependence of the microoperations used in Equations (8)–(11) can be eliminated.

For the LS module, the assembly instructions for Equations (8)–(11) are shown in Algorithm 3.
**Algorithm 3** Assemblyinstruction code for LS estimation.1:**Repeat** N/2 13 {//The next 13 instructions are repeated N/2 times. Four H values of four pilot subcarriers can be obtained for each calculation. Each pilot subcarrier needs to calculate 8 H values in LS estimation. If the number of pilot subcarriers is N, then the number of cycles is N/4 * 2.//G0 stores the R1 and R2 of subcarriers 1-4, G1 and G2 stores the Z1 and Z2, Z3 and Z4 of subcarriers 1-4 respectively.2:8:21CM1C dst [G3:X0-X3], src1[G0:X0-X3]src0[G0:X4-X7];//Divisor for complex division by subcarriers parallelism3:2:2CMASC2C dst [G4:X0-X3], src1[G1:X0-X3] src0[G0:X4-X7];//dividend for complex division by data parallelism4:1CLUT1C dst [G5:X0], src0[G3:X0];//calculating reciprocal by complex division5:1CLUT1C dst [G5:X1], src0[G3:X1];6:1CLUT1C dst [G5:X2], src0[G3:X2];7:1CLUT1C dst [G5:X3], src0[G3:X3];8:2:2CMASC2C dst [G4:X4-X7], src1[G2:X0-X3] src0[G0:X4-X7];9:2:2CMASC2C dst [G4:X8-X11], src1[G2:X0-X3] src0[G0:X4-X7];10:2:2CMASC2C dst [G4:X12-X15], src1[G2:X0-X3] src0[G0:X4-X7];11:8CM1C dst [G6:X0-X3], src1[G5:X0] src0[G4:X0-X3]; // data parallelism, Four H values of one subcarrier per instruction12:8CM1C dst [G7:X0-X3], src1[G5:X1] src0[G4:X4-X7];13:8CM1C dst [G8:X0-X3], src1[G5:X2] src0[G4:X8-X11];14:8CM1C dst [G9:X0-X3], src1[G5:X3] src0[G4:X12-X15];//Four H values of X0-X3 corresponding subcarrier 1 on G6, and subcarrier 2 on G7, and subcarrier 3 on G8, and subcarrier 4 on G9.}

Taking Wiener interpolation in the frequency domain as an example, the assembly instructions of Equations (3) and (4) are shown in Algorithm 4. If linear interpolation in the frequency domain is used, the instruction to calculate the Wiener interpolation tap in lines 6–12 can be replaced by a look-up table of the linear interpolation tap.

### 4.5. Efficiency and Performance Comparison

One of the main pieces of work and innovations of this paper is to design SIMD instructions to accelerate some operations of channel estimation, which greatly speeds up the operation speed of the processor. The instruction set in this paper belongs to the vector operation acceleration instruction set, and one of its instructions can operate on a complex vector group. This means that for the realization of channel estimation functions, the number of instructions used in this paper will be much less than the general instruction set which can only deal with a complex scalar. For the existing general vector processing instruction set, the vector accelerated instruction set proposed in this paper also has significant advantages. The existing general vector processing instruction set can only perform subcarrier parallel operation on the same part of the data of different subcarriers, but not on different data parts of the same subcarrier. In this paper, through the design of subcarrier parallelism, data parallelism and instruction fusion accelerated operation instructions, the instruction efficiency is better than the existing general vector processing instruction set. For example, under the same instruction condition that limits the use of eight complex multipliers, the comparison of the instruction efficiency of each algorithm in the channel estimation body loop using the instruction set of this paper and the general vector processor instruction set, such as the Intel AVX-512 instruction set [26] and Arm Neon instruction set [27], is shown in Table 4. This result is calculated based on the body loop in the assembly code of Algorithms 3 and 4. The number of cycles is N/2 and N/4, respectively, where N represents the number of subcarriers (1024). The VP of this paper is a VLIW load/store and is executed in parallel with SIMD execution, hiding the load/store overhead. In order to make the comparison conditions consistent, we do not consider the load/store operation instructions required by the existing general vector processing instruction set.

As shown in Table 4, after ignoring the load/store operation instructions and promoting SIMD capabilities to 512 bits, the instruction efficiency of the two commercial general vector processing instruction sets is the same. This is because the algorithms of channel estimation can be realized by simple instructions such as multiplication and addition, multiplication and subtraction of vectors and division. These simple instructions are complete in the existing general vector processing instruction set. Therefore, in order to improve the instruction efficiency of the channel estimation module, this study finds the acceleration opportunity and designs the acceleration instruction in some steps of the channel estimation algorithm, which is the main work and contribution of this paper.

The number of instructions in the instruction set proposed in this paper is reduced by eight in time domain Wiener filtering and time domain linear interpolation, because the instruction set proposed in this paper can be accelerated at the data level to use eight complex multipliers for one instruction, and the hardware utilization of the instruction is higher than that of the six complex multipliers in the subcarrier parallel mode of the general vector instruction set. In the linear interpolation algorithm in the frequency domain and the second-order interpolation algorithm in the time domain, the instruction efficiency of the two instruction sets is the same, because the data arrangements of the two algorithms and the algorithm itself are very simple. There is no chance of acceleration from the point of view of data parallelism or subcarrier parallelism. In the second-order interpolation in the frequency domain, because the eight pilot subcarriers are divided into four groups, the general vector processing instruction set that only uses the parallel subcarriers can only use four sets of parallel corresponding complex multipliers on one instruction. In this study, data parallelism is used to make one instruction use six multipliers for computation on the same subcarrier, so the instruction improvement is 32.7%.
**Algorithm 4** Assembly instruction code for Wiener interpolation in frequency domain.// it is necessary to calculate the values of two additional H matrices to make extrapolation estimates before the cycle starts.1:8CM1R dst [G2:X0-X7], src1[G0:X15] src0[G1:X0-X7]; //1st extrapolation,G0:X15 = 22:8CS8C dst [G3:X0-X7], src1[G2:X0-X7] src0[G0:X0-X7];3:8CM1R dst [G6:X0-X7], src1[G0:X15] src0[G5:X0-X7];4:8CS8C dst [G7:X0-X7], src1[G6:X0-X7] src0[G4:X0-X7];5:**Repeat** N/4 8 {// Interpolation loop body. Four H values of 9 pilots are token at a time and interpolate 8 subcarriers.Each subcarrier needs to be interpolated with 8 H values.Then if the total number of subcarriers is N, the number of cycles is 2*N/8.//[G1:X0-X3]: Crosscorrelation matrix generated by look-up table; [G0:X0-X7]:Autocorrelation matrix generated by look-up table; [G1:X8-X11] SNR; [G1:X12]zero. The tap group is calculated once for every four pilot subcarriers, so twice for each RB.6:16CP16C dst [G2:X0-X7], src1[G1:X8,X12,X12,X8,X9,X12,X12,X9] src0[G0:X0-X7];//Calculate the autocorrelation matrix add SNR parameter diagonal matrix then give [a b c d]//Instruction 7–11 is to get the inverse matrix part of two 2 × 2 autocorrelation matrices.7:4:2CMSC2C dst [G3:X0-X1], src1[G2:X2,X3,X6,X7] src0[G2:X0,X1,X4,X5]; //Calculate the determinant ad-bc of the inverse of the autocorrelation matrix. The inverse matrix is obtained every four pilot subcarriers, and the two inverse programs are parallel.8:2:4QY4 dst [G4:X0-X7] src0[G2:X0-X7];//The cofactor of calculating the inverse of autocorrelation matrix [d -b;-c a]9:1RLUT1R dst [G3:X0], src0[G3:X0]; //get 1/(ad-bc)10:1RLUT1R dst [G3:X1], src0[G3:X0];11:2:4CM1C dst [G5:X0-X7], src1[G3:X0,X1] src0[G4:X0-X7];// get the autocorrelation matrix12:2:2MM1 dst [G6:X0-X3], src1[G5:X0-X7] src0[G1:X0-X3];// multiply the autocorrelation and crosscorrelation matrix to get the tap13:Repeat 4 2{//This is the interpolation part. The last two instructions need to be looped four times in this small loop. That is, 8 instructions are required to complete a large cycle of interpolation.// the G7-G10 group is used to store the H values of the nine pilot subcarriers, and the G11-G14 group is used to store the H values of the eight subcarriers to be interpolated. //Each cycle corresponds to an H value of a subcarrier, and four H values of 8 subcarriers are obtained in four cycles.14:Interp dst [G11-G14:X0-X3], src1[G7-G10:X0-X4] src0[G6:X0,X1]; // subcarriers parallelism15:Interp dst [G11-G14:X4-X7], src1[G7-G10:X4-X8] src0[G6:X0,X1];}}

The instruction set proposed in this paper performs well in LS estimation and the Wiener interpolation algorithm in the frequency domain, because one pilot subcarrier needs to be interpolated between every two pilot subcarriers in the frequency domain, and the inverse and interpolation are calculated on the basis of four pilot subcarriers, which makes the data arrangement more complex and gives more opportunities for acceleration. Because of the accelerated instructions in Table 3, compared with the existing general vector processing instruction set, the instruction improvement of the accelerated instruction set proposed in this paper in the LS estimation module is 50%, and that of the Wiener interpolation module in the frequency domain is 37.5%. For each RE, Wiener filter interpolation requires 3/2 more instructions in the frequency domain and 4/3 more instructions in the time domain than linear interpolation, but each gives a BER optimization of 2db, which is worth the effort. Second-order interpolation not only has the maximum number of instructions, but also shows no improvement in BER performance, so it is not used in this study.

## 5. Conclusions

In this study, by designing a set of dedicated instructions on the vector processor, the uplink channel estimation module in 5G small cells is realized. By designing a simple LS estimation algorithm at the pilot point, it can be applied to the pilot structure proposed by 3GPP-38211, and it can complete the channel estimation function at the pilot point with fewer instructions. In this study, the channel estimation algorithm is realized with instruction-level acceleration, including reference signal estimation, Wiener, first-order and second-order frequency domain interpolation and time domain interpolation. After using the look-up table method to store the parameters needed for Wiener filter interpolation in several scenarios, the number of instructions and hardware complexity needed to implement Wiener filtering are reduced. Better results can also be obtained by using it. However, its BER performance is lower than that of the machine learning algorithm. In view of the long convergence time of machine learning, the choice in this study is equivalent to the appropriate tradeoff between low hardware delay and high algorithm performance. Actually, we wanted to find relevant reference works on linear, 2nd-order and Wiener interpolation acceleration on the instruction level. We went through papers but did not find useful references, so we can only learn from commercial instruction sets. We thus hope that our work can be the starter to trigger studies in the field. Instruction-level acceleration is implemented on our vector processors and is also applicable to other commercial and academic vector processors. Acceleration is suitable for both subcarrier parallel and data parallel implementations. Through instruction acceleration, compared with the existing general vector processing instruction set, the processor performance of the LS estimation module is improved by 50%, and the processor performance of the frequency domain Wiener filter interpolation is improved by 37.5%. At the same time, this paper gives the instruction costs of the combination of six interpolation methods. This work can provide a reference for other researchers to study the relationship between the complexity of the algorithm and the number of instructions symbolizing the use of hardware costs. The BER VS SNR measure of time–frequency Wiener filter interpolation achieves 4db compared with linear interpolation, which means that our instruction-level acceleration can be an optimal solution. This paper lacks a discussion on the instruction set for hardware architecture implementation, which will become the research direction in the future.

## Figures and Tables

**Figure 1 sensors-22-00753-f001:**
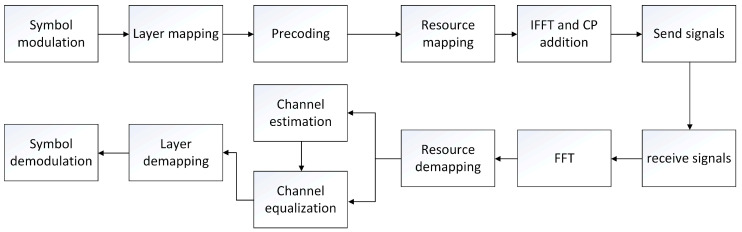
5G flow chart of uplink channel estimation.

**Figure 2 sensors-22-00753-f002:**
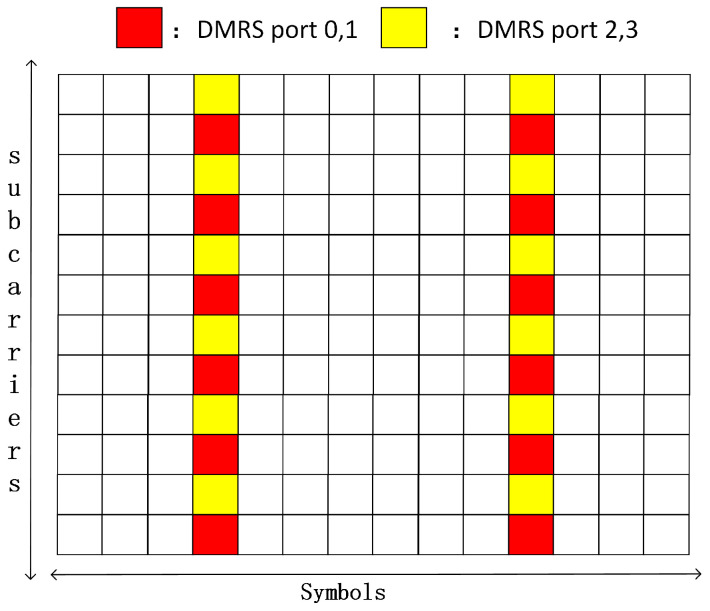
Schematic diagram of single-symbol DMRS pilot type 1.

**Figure 3 sensors-22-00753-f003:**
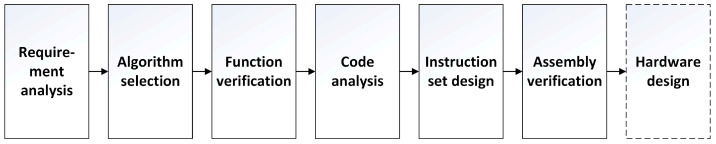
Flow chart of dedicated instruction set design.

**Figure 4 sensors-22-00753-f004:**
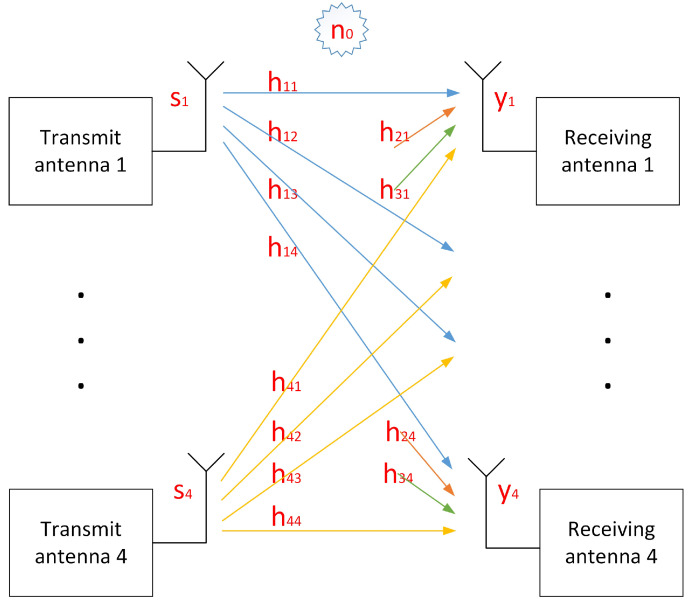
Transmission model of 4 × 4 MIMO system corresponding to Equation (Equation 1).

**Figure 5 sensors-22-00753-f005:**
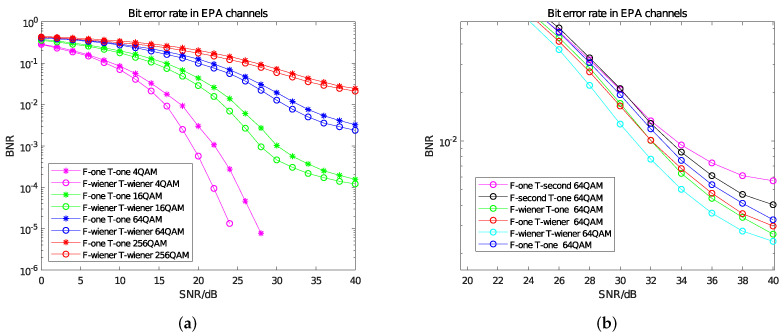
BER curve in EPA channels. (**a**) Comparison of BER curves between Wiener filter interpolation and linear interpolation. (**b**) BER Curves of different interpolation methods under 64 QAM.

**Table 1 sensors-22-00753-t001:** Delay profiles for EPA channel models.

Excess tap delay (ns)	0	30	70	90	110	190	410
Relative power (dB)	0.0	−1.0	−2.0	−3.0	−8.0	−17.2	−20.8

**Table 2 sensors-22-00753-t002:** Relatively general parallel instruction design for SIMD data.

1	2:2CMASC2C	Double parallel instruction of 2 complex vectors multiplied and 2 complex vectors added, and 2 complex vectors multiplied and 2 complex vectors subtracted
2	8CM1C	8 complex vectors multiplied by 1 complex vector and adding values on respective accumulators
3	8CM1R	8 complex vectors multiplied by 1 real scalar
4	8CS8C	8 complex vectors subtract 8 complex vectors
5	2:2MM1	Double parallel instruction of 2 × 2 complex matrix multiplied by 2 complex vectors
6	4:2CMSC2C	Four parallel instructions of 2 complex vectors multiplied and subtract 2 complex vectors
7	16CP16C	16 complex vectors add 16 complex vectors
8	2:4CM1C	Double parallel instruction of 4 complex vectors multiplied by 1 complex vector
9	8CM8C	8 complex vectors multiplied by 8 complex vectors and adding values on respective accumulators
10	1RLUT1R	Using a look-up table to find the reciprocal of a real number

**Table 3 sensors-22-00753-t003:** Dedicated acceleration command for channel estimation module.

11	Interpolation	A group of five continuous DMRSs are multiplied by their corresponding taps to interpolate four subcarriers
12	2:4QY4	Double parallel instruction of finding cofactors in 2 × 2 matrices
13	8:21CM1C	Eight parallel instructions of multiplying a complex number by a complex number and then by 2
14	1CLUT1C	Using two look-up tables for a complex data reciprocal

**Table 4 sensors-22-00753-t004:** Comparison of instruction efficiency for the channel estimation module.

Functional Module	Average Cycles per RE in This Paper	Average Cycles per RE in AVX-512	Average Cycles per RE in Neon	Improvement
LS estimation	26/4 = 6.5	52/4 = 13	52/4 = 13	50%
F-domain Wiener interpolation	30/8 = 3.75	30/8 = 3.75	48/8 = 6	37.5%
T-domain Wiener interpolation	33/6 = 5.5	33/6 = 5.5	41/6 = 6.83	19.5%
F-domain linear interpolation	18/8 = 2.25	18/8 = 2.25	18/8 = 2.25	0
F-domain 2nd-order interpolation	33/8 = 4.125	33/8 = 4.125	49/8 = 6.125	32.7%
T-domain linear interpolation	25/6 = 4.17	25/6 = 4.17	33/6 = 5.5	24.2%
T-domain 2nd-order interpolation	49/6 = 8.17	49/6 = 8.17	49/6 = 8.17	0

## Data Availability

Not applicable.

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
