# Peer review of "An Uplink Channel Estimator Using a Dedicated Instruction Set for 5G Small Cells"

_sensors, 2022, doi:10.3390/s22030753_

Round 1
Reviewer 1 Report
This work deals with a method for channel estimation in 5G small cells. Their goal is to implement an algorithm with easily and based on low complexity. The authors claim that they achieve this goal with interpolation using the Wiener model. I personally struggled to understand where exactly the proposed method fits into the 5G system as a whole. I recommend improvements to the introduction, methodology description, measurements and conclusion. In my opinion this is a scientific work that has merit and potential results but needs to be better presented. For example, see that the work has only 12 references, many of them from conferences that do not have the same expression as journal papers.
1) In the introduction the contextualization is very poor. The way it is currently being done is based on articles from almost a decade ago (Refs 2-4) and which cannot be considered recent in this field of research, despite being relevant. Below there is one article that should be cited in the scope of this research. I am including two others that the authors should also use as example of the relevance and constrains of their method. Anyway I recommend a significant improvement in the introduction section.
Shahabuddin, S., Mämmelä, A., Juntti, M., & Silvén, O. (2021). ASIP for 5G and Beyond: Opportunities and Vision. IEEE Transactions on Circuits and Systems II: Express Briefs, 68(3), 851-857.
Liu, S., & Liu, D. (2018). A high-flexible low-latency memory-based FFT processor for 4G, WLAN, and future 5G. IEEE Transactions on Very Large Scale Integration (VLSI) Systems, 27(3), 511-52
Khan, I., & Singh, D. (2018). Efficient compressive sensing based sparse channel estimation for 5G massive MIMO systems. AEU-International Journal of Electronics and Communications, 89, 181-190.
2) What is the reference source of Fig1?
3) Section 2 which is actually where you will present your model is very short and very succinctly describes the problem. I would recommend improving this section by enriching the model description.
4) Please include the proper references for the used equations (1,2,3,5,6);
5) A Figure would be recommended for explaining the example of paragraph of line 97 and Eq(1);
6) References [8] and [10] they deserved a more detailed explanation when they were cited in the manuscript text. As it is, we don't have a reasonable idea of what would actually be used/applied. I recommend a more detailed explanation in this regard.
7) Line 152, please justify the 410 ns value;
8) It would be very important to cite other results about this issue addressed in the literature. The results are presented with expressive gains, but I missed a discussion of these results with others who worked on the same theme.
9) Please explain what the term N in the algorithm is.
10) The conclusion needs to be better written with more details of the benefits and limitations of the proposed method.
Reviewer 2 Report
There are many issues in this paper. Notwithstanding, all of them are addressable, as follows:
- Please select the keywords carefully by including only those which have been used in the title and abstract.
- The introduction section is short and does not include the list of contributions.
- Some of the other sections are also short, for example 2.1. So, please make sure those sections have consistent length with others.
- The formal modelling is good, though the conceptual model is poor.
- The communication part over 5G requires more support and references. There are many recent research papers in this area, such as: https://ieeexplore.ieee.org/abstract/document/7930654 so, please add more to address this comment.
- Some of the algorithms are poorly written and explained, e.g. Algorithm 4. Please make sure they can be easily read and discussed.
- The conclusion is currently like a summary, you need to express the lessons learned and future directions.
- The number of references is very little. This implies that the authors have not done a good survey on the literature, hence it affects the reliability of the presented work. Please conduct more literature study and add more references.
Reviewer 3 Report
- Results: Recommend to be Major revisions
This paper implements channel estimation algorithms though instruction level acceleration including reference signal estimation, Wiener, 1st order, and 2nd order interpolations in frequency and time domains. The instruction level acceleration is on our vector processor, and is yet suitable for other commercial & academic vector processors. Accelerations are for both sub-carrier parallel and data parallel implementations. Through instruction acceleration, compared with the existing general vector processing instruction sets, the processor performance of the LS estimation module is improved by 50% and the processor performance of Wiener filter interpolation in frequency domain is improved by 37.5%. The BER VS SNR measure of time-frequency Wiener filter interpolation has 4db achievement compared with linear interpolation, meaning our instruction level acceleration can be of an optimum solution.
This paper is with some merits for Sensors, thus, it requires some major revisions.
Firstly, the abstract should be refined to clearly indicate what authors had done within 150 words.
Secondly, for Section 1, authors should provide the comments of the cited papers after introducing each relevant work. What readers require is, by convinced literature review, to understand the clear thinking/consideration why the proposed approach can reach more convinced results. This is the very contribution from authors. In addition, authors also should provide more sufficient critical literature review to indicate the drawbacks of existed approaches, then, well define the main stream of research direction, how did those previous studies perform? Employ which methodologies? Which problem still requires to be solved? Why is the proposed approach suitable to be used to solve the critical problem? We need more convinced literature reviews to indicate clearly the state-of-the-art development. And very importantly, authors always have to write a paragraph saying: “The rest of the paper is organized as follows. Section 2 contains the literature review. Section 3 contains the methodology (method). Section 4 contains the results. Section 5 contains the conclusions and policy implications”. So, the reader knows what’s coming next. By the way, several listed references, such as [12], does not appear in the text, please check them carefully.
For Section 2, authors should introduce their proposed research framework more effective, i.e., some essential brief explanation vis-à-vis the text with a total research flowchart or framework diagram for each proposed algorithm to indicate how these employed models are working to receive the experimental results. It is difficult to understand how the proposed approaches are working.
For Sections 3 and 4, authors should use more alternative models as the benchmarking models, authors should also conduct some statistical test to ensure the superiority of the proposed approach, i.e., how could authors ensure that their results are superior to others? Meanwhile, authors also have to provide some insight discussion of the results. Authors can refer the following references for conducting statistical test.
Forecasting short-term electricity load using hybrid support vector regression with grey catastrophe and random forest modeling. Utilities Policy, 2021, 73, 101294.
Round 2
Reviewer 1 Report
The requested adjustments in the text have been made. This work is modest but sounds ok
Reviewer 2 Report
The authors addressed all comments and the paper is ready for publication.
Reviewer 3 Report
Authors have completely addressed all my concerns.